# Effect of Data-Processing Methods on Acceleration Summary Metrics of GNSS Devices in Elite Australian Football

**DOI:** 10.3390/s24134383

**Published:** 2024-07-05

**Authors:** Susanne Ellens, David L. Carey, Paul B. Gastin, Matthew C. Varley

**Affiliations:** Sport, Performance, and Nutrition Research Group, School of Allied Health, Human Services and Sport, La Trobe University, Melbourne 3000, Australia; s.ellens@latrobe.edu.au (S.E.); d.carey@latrobe.edu.au (D.L.C.); p.gastin@latrobe.edu.au (P.B.G.)

**Keywords:** data processing, smoothing, filter, GPS, acceleration

## Abstract

This study aimed to measure the differences in commonly used summary acceleration metrics during elite Australian football games under three different data processing protocols (raw, custom-processed, manufacturer-processed). Estimates of distance, speed and acceleration were collected with a 10-Hz GNSS tracking technology device from fourteen matches of 38 elite Australian football players from one team. Raw and manufacturer-processed data were exported from respective proprietary software and two common summary acceleration metrics (number of efforts and distance within medium/high-intensity zone) were calculated for the three processing methods. To estimate the effect of the three different data processing methods on the summary metrics, linear mixed models were used. The main findings demonstrated that there were substantial differences between the three processing methods; the manufacturer-processed acceleration data had the lowest reported distance (up to 184 times lower) and efforts (up to 89 times lower), followed by the custom-processed distance (up to 3.3 times lower) and efforts (up to 4.3 times lower), where raw data had the highest reported distance and efforts. The results indicated that different processing methods changed the metric output and in turn alters the quantification of the demands of a sport (volume, intensity and frequency of the metrics). Coaches, practitioners and researchers need to understand that various processing methods alter the summary metrics of acceleration data. By being informed about how these metrics are affected by processing methods, they can better interpret the data available and effectively tailor their training programs to match the demands of competition.

## 1. Introduction

Global navigation satellite systems (GNSS) are a commonly used athlete tracking system in team sports and permit the quantification of player movement [1,2]. A GNSS device accesses satellites from multiple constellations in orbit (e.g., GPS and GLONASS) to determine its position in space, allowing the estimation of its distance covered, speed and acceleration [3,4]. In addition, some athlete tracking systems also include a triaxial accelerometer, gyroscope and magnetometer, allowing human activity recognition and the measurement of variables such as PlayerLoad™ [5,6]. The accelerometers within the athlete tracking system are not involved in the calculation of GNSS acceleration; accelerometer-derived acceleration is distinctly different data. Notably, most researchers and practitioners (79%) in team sports use GNSS-derived acceleration data [7]. Accurately quantifying player movements by determining the intensity, frequency and volume of these movements demonstrates the demands of a sport [8]. This knowledge can be used by practitioners to design training programs that adequately prepare athletes for competition [8].

Acceleration-based movements have been highlighted across the literature as important for team sport performance [9], as many movements require an athlete to accelerate or decelerate (negative acceleration) rapidly. Within sports, GNSS time-series acceleration data are often summarised according to the distance run or number of efforts performed within certain acceleration/deceleration bands [1,7,9]. Acceleration efforts have been identified as a critical component of Australian rules football (AF), where acceleration match demands increase with increasing competition level [10]. Therefore, quantifying these acceleration-based movements in AF are of interest to coaches and practitioners. Manufacturer data processing can have a large influence on GNSS acceleration data and corresponding summary metrics such as number of efforts [11,12]. Different data processes can alter the quantification of player movement, which may affect a practitioner’s interpretation of the data and training programs. For example, applying a data processing method with strong smoothing, can cause a reduction in the number of acceleration efforts recorded during a match [11], potentially changing a practitioners interpretation of players’ workload. Valid GNSS acceleration data are needed to correctly quantify player movement with summary metrics.

Various validation studies have assessed the ability of GNSS to estimate acceleration [12,13,14,15,16]. This is an ongoing process as each new GNSS device requires validation [17]. Numerous GNSS manufacturers apply their own data processing methods within their software, which could impact data validity and result in manufacturer-influenced variations in summary metrics, rather than being directly related to what the GNSS device is measuring [17]. Within the literature, large manufacturer-influenced variations have been reported in summary acceleration metrics [11,18]. Variations of up to ~250 acceleration efforts have been observed when using different manufacturer data processing methods on an identical dataset from a soccer match [11]. Although it is known that differences exist between manufacturer software processed data, most practitioners still use manufacturer’s software-derived GNSS data, as it is a simple and efficient way to obtain data.

Practitioners and researchers experienced with data processing techniques might choose to extract and process the raw (not smoothed in any way by the manufacturer software) data from the GNSS devices and analyse it separately [9,19]. This approach offers several advantages such as eliminating undesired processing practices (e.g., smoothing and algorithms) and incorporating custom processes such as new summary metrics [9,17]. Custom processing of manufacturer-exported GNSS data has been shown to enhance acceleration data quality, and derived summary metrics differed compared to manufacturer-processed data [20]. When using custom processing on manufacturer-processed GNSS data, double processing and over-smoothing of the data could take place, which could eliminate important parts of the acceleration data. Using custom processing on raw GNSS data would eliminate double processing and could enhance data quality. However, there is no research comparing custom-processed raw GNSS data to manufacturer-processed data.

While the three methods (raw, custom-processed and manufacturer-processed) are available for analysis of GNSS data, there is no research identifying differences in the summary acceleration metrics between the methods. Furthermore, there is limited research exploring differences between summary acceleration metrics of just the raw and manufacturer-processed data. Large differences have been reported in the distance covered when accelerating between raw and manufacturer-processed data in a controlled environment where GNSS devices were positioned on a sled [18], and large differences have been found between just the acceleration data of team sport training sessions [20]. However, no study has investigated the difference in commonly used acceleration metrics between raw and manufacturer-processed data of team sport players during competition match-play.

Therefore, the aim of this study was to compare and explore the differences between three data processing methods (I. raw; II. custom-processed; III. manufacturer-processed) in the commonly used acceleration metrics of elite Australian football competition match-play data using GNSS tracking technology. To make this research practical, it was decided that GNSS-derived acceleration data would be used rather than acceleration data measured by an accelerometer, as most researchers and practitioners in team sports use GNSS-derived acceleration data.

## 2. Methods

### 2.1. Participants

Player movement data from thirty-eight elite male players from one Australian Football League team were collected with a 10-Hz GNSS tracking technology device from fourteen matches during the 2019 competitive season. Data were included if the horizontal dilution of precision (HDOP) was ≤2 and the player was in play on the field. There was no minimum playing duration requirement for a player file to be included. This resulted in a total sample size of 262 player files. The procedures used in this study were conducted with approval from the Human Research Ethics Committee of La Trobe University (reference number: HEC21282).

### 2.2. Equipment

Player movement, including estimations of acceleration, speed and distance, was measured using a 10-Hz GNSS device (Vector S7, Catapult Innovations, Melbourne, Australia). The device was positioned between the athlete’s shoulder blades using the manufacturer’s snug-fitting garment to prevent unnecessary device movement. Data collection procedures adhered to the guidelines outlined by Malone, Lovell, Varley and Coutts [17], with each athlete having their own specific device. The study sample had an average (±SD), 12 ± 1 number of satellites and horizontal dilution of precision (HDOP) of 0.62 ± 0.07.

The Vector S7 has reported acceptable levels of reliability and validity for speed (coefficient of variation ≤2%, mean bias −0.5%) and reliability and validity for distance (coefficient of variation ≤1.3%, mean bias ≤1%) according to Catapult’s vector data integrity testing [21] and peer reviewed research [22].

The raw (not smoothed in any way by the manufacturer software) GNSS Doppler-shift speed data were exported and retrieved from the Catapult software (Openfield, version 2.7.1, Catapult Sports, Melbourne, Australia) files folder. The raw acceleration dataset was calculated using a central difference method on the raw Doppler-shift speed data. To determine the most appropriate custom processing method, several common smoothing methods (Butterworth filter: cut-off frequencies 0.1 to 4.9 Hz, exponential smoothing: smoothing constant 0.1 to 0.9, moving average: sliding window 0.1 s to 0.9 s) have been applied to the raw GNSS Doppler-shift speed data and were compared with a gold standard motion analysis system (Vicon) dataset. The fourth order (zero lag) low-pass Butterworth filter with a cut-off frequency of 2 Hz showed the strongest relationship with the Vicon data (mean bias 0.00 m·s^−2^, 95% LoA ± 1.55 m·s^−2^, RMSE 0.79 m·s^−2^) and was therefore used on the raw GNSS Doppler-shift speed data for the custom processing method. After using the Butterworth filter, acceleration was calculated using a central difference method on the custom-processed GNSS Doppler-shift speed data. Applying the processing to the raw Doppler-shift speed data before deriving acceleration data will ensure that any noise present in the raw Doppler-shift speed data will not be increased due to deriving acceleration. The manufacturer-processed GNSS distance and acceleration data were exported from the manufacturer’s software using their default settings (Openfield, version 2.7.1, Catapult Sports, Melbourne, Australia). A summary of the details of the three datasets used for further analysis, (I) raw, (II) custom-processed, (III) manufacturer-processed, can be found in Table 1.

As the custom-processed data were derived from the raw data, these datasets were automatically synchronised. To allow for comparison of all results, the manufacturer-processed data were synchronised with the raw data. The raw data files represented all data from the time the GNSS units were switched on to start data collection until they were switched off. However, the manufacturer-processed data represented only gametime. Consequently, the files varied in length and could not be synchronised by means of cross-correlation. Unix timestamps (also known as epoch time) were used for synchronisation. The raw data files only included one Unix timestamp corresponding to the time the GNSS units were switched on to start data collection; they did not include a timestamp variable. The Unix timestamp was used to create a timeseries for the raw datafiles by extending the Unix timestamp by the sampling frequency of the device and the length of each file. The Unix timeseries of the raw and manufacturer-processed datasets were then used to synchronise and join both datasets. Cross-correlation analysis was performed afterwards to confirm perfect alignment of the raw and manufacturer-processed data.

### 2.3. Data Analysis

Two common summary acceleration metrics were extracted from the datasets. The first metric was the number of high- and medium-intensity acceleration and deceleration efforts which were extracted for each player and game from each of the three acceleration datasets. The start of a high effort was defined by a ±3 m·s^−2^ threshold (a negative threshold defines a deceleration effort and a positive threshold an acceleration effort) and ±2 m·s^−2^ for medium efforts. These thresholds were selected as they are commonly used for high and medium acceleration and deceleration efforts in the research literature [7,9,23]. An effort was counted when the acceleration data reached the set threshold and stayed above the set threshold for at least 0.3 s [24] and ended when the acceleration data reached 0 m·s^−2^ [11]. The second metric was the distance covered in meters using the manufacturer-processed GNSS distance within a predefined high (≥3 m·s^−2^ for acceleration and ≤−3 m·s^−2^ for deceleration) and medium (2 to 3 m·s^−2^ for acceleration and −2 to −3 m·s^−2^ for deceleration) intensity zone, extracted from each of the three acceleration datasets.

### 2.4. Statistical Analysis

To estimate the effect of the three different data processing methods on the number of acceleration/deceleration efforts (#) and distance (m) within the high-intensity zone (±3 m·s^−2^) or medium-intensity zone (±2 m·s^−2^), linear mixed models were used to account for recurring measures. A negative binomial generalised linear mixed model was used for the effort model, as the efforts were count-based, not normally distributed data [25], and a linear mixed model was used for the distance model [26]. Each model included a fixed effect for processing method (raw, custom-processed, manufacturer-processed). The models included a random effect for player ID and game, which allowed for different mean values for each player and game. The change in number of efforts or distance reported between processing methods within the medium- or high-acceleration or -deceleration zone was estimated, and a 95% confidence interval (CI) was used to denote the imprecision of the fixed effect parameter estimates. To determine the difference in number of efforts (#) and distance (m) within each processing method between the high-intensity zone or medium-intensity zone, the same mixed models were used as mentioned above, but with intensity (medium or high) as a fixed effect. All analysis were performed in MATLAB (version 9.14.0 (R2023a), The MathWorks Inc., Natick, MA, USA).

## 3. Results

### 3.1. Between Processing Methods Effects

Overall, manufacturer processing had the lowest reported distance and efforts, followed by the custom processing, then the raw data. When using manufacturer-processed data, distance covered while accelerating above the high threshold was on average 7.5 m, whereas the custom processing brought the distance up to an average of 421 m and the raw data, 1380 m (Figure 1). For the efforts reported in the high-acceleration zone, manufacturer processing reported 3 efforts on average, where custom processing reported 138 efforts and raw data 224 efforts.

For the variable distance in meters while in the high deceleration zone, the main effect for processing method was significant (F(2, 696) = 5371, *p* < 0.001). Comparing the custom processing to manufacturer processing, distance increased by 217 m, (95% confidence interval (CI) = [200 m to 233 m], t(696) = 25, *p* < 0.001). Comparing the raw data to manufacturer processing, distance increased by 849 m, (95% CI = [832 m to 865 m], t(696) = 100, *p* < 0.001). Comparing the raw data to custom processing, distance increased by 632 m, (95% CI = [615 m to 649 m], t(696) = 74, *p* < 0.001). The results of the effect of processing method on distance in the medium/high-acceleration or -deceleration zones are presented in Table 2.

For the variable number of efforts while in the high-deceleration zone, the main effect for processing method was significant, X^2^(2) = 15,938, *p* < 0.001. Comparing the custom processing to manufacturer processing, the number of efforts increased 8.88 times, (95% CI = [8.51 to 9.57], *p* < 0.001). Comparing the raw data to manufacturer processing, the number of efforts increased 15.2 times, (95% CI = [14.6 to 15.9], *p* < 0.001). Comparing the raw data to custom processing, the number of efforts increased 1.71 times, (95% CI = [1.67 to 1.77], *p* < 0.001). The results of the effect of processing method on number of efforts in the medium/high acceleration or deceleration zones are presented in Table 3.

### 3.2. Within Processing Method Effects

Overall, the distance while accelerating or decelerating was largest in the medium zone compared to the high zone for all processing methods, except for distance while accelerating for the raw method. The number of efforts was largest in the high zone compared to the medium zone for all processing methods except manufacturer processing.

For the variable distance while decelerating processed by the manufacturer, the main effect for intensity was significant (F(1, 453) = 4345, *p* < 0.001). Distance in the medium-intensity zone increased by 79 m compared to the high-intensity zone (95% CI = [77 m to 81 m], t(453) = 66, *p* < 0.001). The results of the effect of intensity on distance while decelerating or accelerating for each processing method are presented in Table 4.

For the variable number of efforts while decelerating processed by the manufacturer, the main effect for intensity was significant (X^2^(2) = 1368, *p* < 0.001). The number of efforts in the medium-intensity zone was 2.28 times greater compared to the high-intensity zone (95% CI = [2.18 to 2.38], *p* < 0.001). The results of the effect of intensity on number of efforts while decelerating or accelerating for each processing method are presented in Table 5.

## 4. Discussion

This study aimed to measure the differences in commonly used summary acceleration metrics during elite Australian football games of GNSS acceleration data that were derived using three different processing methods (raw, custom-processed, manufacturer-processed). The main finding was that there were substantial differences between the three processing methods when calculating the same metric. Overall, compared to the raw data, the manufacturer-processed acceleration data had the lowest reported distance (up to 184 times lower) and efforts (up to 89 times lower), followed by the custom-processed distance (up to 3.3 times lower) and efforts (up to 4.3 times lower), where raw data had the highest reported distance and efforts.

The raw data were unprocessed and consequently had the most noise present, resulting in the highest distance covered and number of efforts. The manufacturer-processed acceleration data had the lowest reported distance and efforts. The results were approximately 28 efforts lower than those found in literature using a similar GNSS device and manufacturer software [27]. The difference could be explained by the fact that Rennie, Kelly, Bush, Spurrs, Austin and Watsford [27] used a lower threshold of ±2.78 m·s^−2^ and a shorter duration above the set threshold of 0.2 s to identify an effort. A lower duration above the threshold of 0.2 s vs. 0.3 s (which was used in this study) has been shown to identify 45% more acceleration efforts and 13% more deceleration efforts [24]. Furthermore, all data that involved <75% of total game time were excluded from Rennie, Kelly, Bush, Spurrs, Austin and Watsford [27], while there was no minimum game time requirements for this study. When taking all factors (lower threshold, shorter duration above the threshold and game time criteria) into consideration, the number of efforts reported are comparable to the current study. This further highlights the effects of different processing methods on acceleration data and accompanying difficulty in comparing results across studies.

Coaches and practitioners use acceleration metrics to quantify the demands (volume, intensity and frequency of the metrics) of a sport or activity [7,28], which can be used to create training programs to adequately prepare players for competition [8]. Furthermore, researchers could be using and analysing GNSS manufacturer-processed acceleration data and metrics for their research. A change in the metric output due to processing methods will alter the demands they are quantifying. For example, a sudden increase in the volume of acceleration undertaken might indicate to a coach that players are working harder than normal. Although they are undertaking the same amount of work as usual, the metric output increased due to different processing methods used. Examples of when processing methods might change include when software is updated, athlete tracking technologies are changed or when different software/algorithms are used to process the data. Coaches, practitioners and researchers should be aware that processing methods can change, and that these changes could affect metric outputs and alter the demands they are quantifying.

To be able to select a suitable processing technique for acceleration data, one should be aware of the characteristics of their data (patterns and frequencies that could be present in the data). The characteristics of the data can determine what type of processing method is most suitable [29]. Based on the results of this study, practitioners using GNSS acceleration data are recommended to select a processing method specific to their use case and characteristics of the data. It is also recommended to evaluate the impact of different processing methods on metrics of interest (e.g., number of efforts). For this study, the characteristics of team sport-specific human movement patterns (elite AF players) and potential sources of variability in the acceleration data should be taken into consideration. An athlete in full sprint could have anywhere between 2 and 5 steps per second [30], indicating that at least 2 Hz patterns (corresponding to 2 steps per second) could be present in the acceleration data. The acceleration data varies within a single step (from the heel strike of one foot to the subsequent heel strike of the other foot), which is a result of changing the balance between the propulsive and braking forces at each ground contact [31]. A surplus in propulsive forces results in acceleration, where a surplus of braking forces results in deceleration. If an athlete performs 2 steps per second, that means the acceleration data change significantly at each single step [32]. This suggests that acceleration data might be more variable and higher than what is currently indicated by GNSS devices.

The high acceleration values (considered as acceleration and deceleration values with a high rate of change in speed) are shown by the results of the custom processing method, which was smoothed with a filter which had a 2 Hz cutoff frequency, allowing for 2 Hz patterns in the data. This method might be the closest approximation to the real world of all three processing methods. The larger distance run and number of efforts in raw and custom-processed data compared to manufacturer-processed data is an indicator of stronger data smoothing in the manufacturer-processed data. The custom-processed and raw dataset showed that the acceleration data exceeded the high threshold (±3 m·s^−2^) for more than 0.3 s, significantly more (up to 89.7 times) than the manufacturer-processed dataset (see Table 3). Strong smoothing in the manufacturer data could eliminate important portions of a signal by smoothing peaks and lowering the amplitude of high acceleration data. The amplitude of the manufacturer-processed acceleration data is lowered to the point where more data exceed the medium threshold (±2 m·s^−2^) but stay below the high threshold (±3 m·s^−2^), as evident by the larger number of efforts in the medium zone compared to the high zone (see Table 5). When looking into the literature, athletes have reached acceleration values between 5–7 m·s^−2^ [32,33,34,35], suggesting that elite AF players should be able to reach these values. However, the manufacturer-processed data suggest that the elite AF players barely reach the set ±3 m·s^−2^ threshold, which is an indication that manufacturer-processed data may be over-smoothing and masking the actual acceleration values that an athlete is capable of.

The application of a data processing method with strong smoothing has been shown to cause a reduction in the number of recorded acceleration efforts during a match [11]. Furthermore, manufacturer-processed GNSS acceleration data have shown a very large mean bias, with lower acceleration values, when compared to a criterion measure [12]. In combination with the results from the current study, these findings collectively suggest that manufacturer-processed data are subject to extensive data smoothing.

Distance run in the medium-acceleration zone exceeded that of the high zone for all different processing techniques except for the raw acceleration dataset, where the distance recorded in the high zone was greater. The raw acceleration dataset was not subject to smoothing, meaning that all potential sources of noise, such as sensor movement, multipath interference and environmental conditions [36], were present in the data. This noise may manifest as high-frequency components, leading to a larger representation of raw data in the high zone compared to the medium zone and all other processing methods. Similar findings have been reported for distance based summary acceleration metrics between raw and manufacturer-processed data in a controlled environment [18] and between the acceleration data of team sport training sessions [20].

Large inconsistencies exist in the literature for reported processing steps used on acceleration data, which hinders the comparison of acceleration summary metric results between studies [7,37]. The findings of the current study demonstrated substantial differences between different processing methods when estimating the same acceleration and deceleration metric. Therefore, future research should report all different processing steps performed on their used acceleration data derived from an athlete tracking system to ensure comparability of results between studies.

A noteworthy strength of this research is the use of elite Australian Football team data. The dataset consisted of games played on different days and at different locations within stadia, providing a real representation of diverse GNSS team data. The data were collected with one type of GNSS device and is thus only representative of this specific device. Furthermore, the manufacturer-processed data were exported using the specific manufacturer software mentioned in the methods section. Since data processing procedures could vary between software versions, it is important to note that the manufacturer-processed data are representative only of this specific software. Future research investigating processing methods of acceleration data of athlete tracking technologies should consider using local positioning systems (LPS) and optical positioning systems, next to GNSS. Current elite team sport environments require teams to use a variety of athlete tracking technologies suitable for different locations, e.g., LPS or optical for indoor stadia, GNSS for outdoor or training sessions [18]. The use of different athlete tracking systems interchangeably, requires research to establish the influence of data processing on acceleration data of different tracking systems to be able to compare acceleration data longitudinally.

## 5. Conclusions

The results from this study demonstrated that there were substantial differences in commonly used summary acceleration metrics (number of efforts performed and distance covered) during elite Australian football games between three processing methods (raw, custom-processed, manufacturer-processed). Overall, the manufacturer-processed acceleration data had the lowest reported distance and efforts, followed by the custom-processed distance, where raw data had the highest reported distance and efforts. The results indicate that different processing methods changed the metric output (number of efforts and distance covered) and can in turn alter the quantification of the demands of a sport (volume, intensity and frequency of the metrics). It is important for coaches, practitioners and researchers using GNSS-derived acceleration data to know how, and be aware that, processing methods change summary acceleration metrics (e.g., efforts and distance covered) because they are often used to quantify the demands of a sport and to create training programs to adequately prepare players for competition. Furthermore, it is recommended that future research and tracking technology manufacturers report all data processing practises performed on the acceleration data where possible. Knowing all performed processing steps allow for comparability of results and the ability to identify if differences may be due to processing practises rather than the used tracking technology.

## Figures and Tables

**Figure 1 sensors-24-04383-f001:**
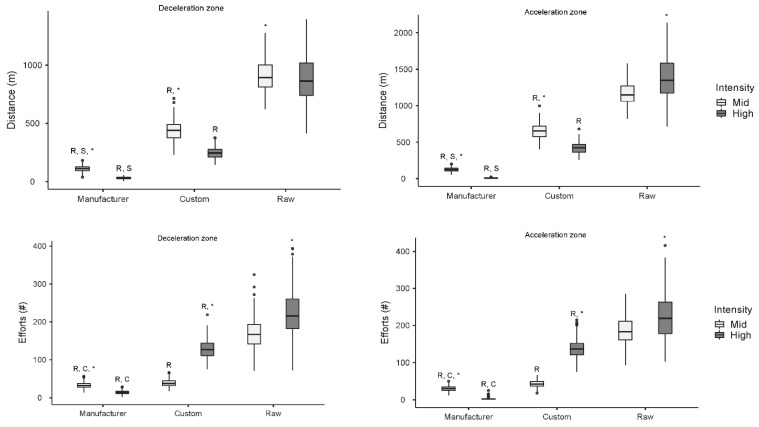
Distance (m) decelerating (**left–top**) and accelerating (**right–top**), number of deceleration efforts (**left–bottom**) and acceleration efforts (**right–bottom**), calculated for Medium (±2 m·s^−2^) and High (±3 m·s^−2^) zones for one dataset processed in three different ways (I) raw, (II) custom-processed, (III) manufacturer-processed. R = Data are different compared to Raw dataset of the same intensity. C = Data are different compared to custom-processed dataset of the same intensity. * = Significantly higher distance compared to the other intensity which was processed the same way.

**Table 1 sensors-24-04383-t001:** Summary of the three used data processing methods and details on how acceleration data were obtained for each processing method.

Data Processing Method	How the Acceleration Data Were Obtained
Raw	Central difference method applied to raw GNSS Doppler-shift speed data to calculate acceleration.
Custom	Raw GNSS Doppler-shift speed data were processed with a fourth order (zero lag) low-pass Butterworth filter with a cut-off frequency of 2 Hz, whereafter acceleration was calculated using a central difference method on the processed GNSS Doppler-shift speed data.
Manufacturer	GNSS acceleration data were directly exported from manufacturer software using their default settings.

**Table 2 sensors-24-04383-t002:** Effects of processing method (raw, custom-processed, manufacturer-processed), on distance in meters split by medium (±2 m·s^−2^)- and high (±3 m·s^−2^)-acceleration or -deceleration intensity.

	Effect of Processing Method on Distance							
	Intensity	Effect	Estimate (m)	Lower 95% CI	Higher 95% CI	df	t	*p*
Acceleration	High	Custom–Manufacturer	413	389	437	697	34	<0.001
		Raw–Manufacturer	1373	1349	1397	697	112	<0.001
		Raw–Custom	959	935	983	696	78	<0.001
	Medium	Custom–Manufacturer	529	515	543	719	74	<0.001
		Raw–Manufacturer	1042	1028	1056	719	145	<0.001
		Raw–Custom	513	499	527	719	71	<0.001
Deceleration	High	Custom–Manufacturer	217	200	233	696	25	<0.001
		Raw–Manufacturer	849	832	865	696	100	<0.001
		Raw–Custom	632	615	649	696	74	<0.001
	Medium	Custom–Manufacturer	327	315	339	719	53	<0.001
		Raw–Manufacturer	798	786	810	719	130	<0.001
		Raw–Custom	471	459	484	719	76	<0.001

df = degrees of freedom; t = t-statistic; *p* = *p*-value.

**Table 3 sensors-24-04383-t003:** Effects of processing method (raw, custom-processed, manufacturer-processed), on number of efforts split by medium (±2 m·s^−2^) and high (±3 m·s^−2^) acceleration or deceleration intensity.

Effect of Processing Method on Number of Efforts						
	Intensity	Effect	Estimate (Rate of Change)	Lower 95% CI	Higher 95% CI	*p*
Acceleration	High	Custom–Manufacturer	55.7	51.3	60.4	<0.001
		Raw–Manufacturer	89.7	82.6	97.4	<0.001
		Raw–Custom	1.61	1.57	1.66	<0.001
	Medium	Custom–Manufacturer	1.43	1.38	1.48	<0.001
		Raw–Manufacturer	6.13	5.93	6.34	<0.001
		Raw–Custom	4.29	4.16	4.43	<0.001
Deceleration	High	Custom–Manufacturer	8.88	8.51	9.57	<0.001
		Raw–Manufacturer	15.2	14.6	15.9	<0.001
		Raw–Custom	1.71	1.67	1.77	<0.001
	Medium	Custom–Manufacturer	1.18	1.14	1.23	<0.001
		Raw–Manufacturer	5.09	4.92	5.27	<0.001
		Raw–Custom	4.30	4.16	4.45	<0.001

**Table 4 sensors-24-04383-t004:** Effects of acceleration or deceleration intensity, medium (±2 m·s^−2^) and high (±3 m·s^−2^), on distance in meters split by processing method (raw, custom-processed, manufacturer-processed).

Effect of Intensity on Distance								
	Processing	Effect	Estimate (m)	Lower 95% CI	Higher 95% CI	df	t	*p*
Acceleration	Manufacturer	Medium–High	118	114	121	455	74	<0.001
	Custom	Medium–High	234	225	243	455	51	<0.001
	Raw	Medium–High	−211	−234	−189	455	−18	<0.001
Deceleration	Manufacturer	Medium–High	79	77	81	453	66	<0.001
	Custom	Medium–High	190	182	197	455	52	<0.001
	Raw	Medium–High	29	13	45	455	3.4	<0.001

df = degrees of freedom; t = t-statistic; *p* = *p*-value.

**Table 5 sensors-24-04383-t005:** Effects of acceleration or deceleration intensity, medium (±2 m·s^−2^) and high (±3 m·s^−2^) on number of efforts, split by processing method (raw, custom-processed, manufacturer-processed).

Effect of Intensity on Number of Efforts						
	Processing	Effect	Estimate (Rate of Change)	Lower 95% CI	Higher 95% CI	*p*
Acceleration	Manufacturer	Medium–High	12.1	11.1	13.1	<0.001
	Custom	Medium–High	0.71	0.63	0.79	<0.001
	Raw	Medium–High	0.83	0.81	0.85	<0.001
Deceleration	Manufacturer	Medium–High	2.28	2.18	2.38	<0.001
	Custom	Medium–High	0.30	0.29	0.31	<0.001
	Raw	Medium–High	0.66	0.60	0.72	<0.001

## Data Availability

The data presented in this study are available on request from the corresponding author. The data are not publicly available due to ethical restrictions.

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
