# Peer review of "Effect of Data-Processing Methods on Acceleration Summary Metrics of GNSS Devices in Elite Australian Football"

_sensors, 2024, doi:10.3390/s24134383_

Round 1

Reviewer 1 Report

Comments and Suggestions for Authors

Those are mine remarks and questions pointing out the weaknesses of the manuscript.

27: GPS or GNSS?

405-406: I am not sure if a publication submitted for publication may be used as a valid reference.

99-102: Mentioning of new terms and techniques by referencing to a paper submitted for publication seems unacceptable.

115: According to specifications of the Vector S7 device (available at https://support.catapultsports.com/hc/en-us/article_attachments/360007134595) it is about 'an athlete tracking device with the capability of capturing global (GPS) and local (LPS) positioning information, inertial and heart rate data'. Therefore, presenting a tracking device as GNSS device could lead to a misleading information. I suggest to remove the term 'GNSS' from the title, because the paper itself has a little to do with GNSS and by this creating a misleading situation for the reader.

122: What is the meaning of 'CV'?

129-130: Table 1 gives very few information about processing methods, their specification and algorithm properties. Moreover, the reader is advised to look for additional information in an unpublished paper.

In section '2.3 Data Analysis': the key question is how acceleration data was obtained: by GNSS observations or by accelerometer measurements??? If the acceleration was obtained by accelerometer measurements, what the acceleration data has to do with GNSS??? If the acceleration was derived by GNSS observations, there is no word about methodology.

163-164: Additional information and appropriate references related to 'A negative binomial generalized linear mixed model, and a linear mixed model' are missing.

What is the meaning of 'CI' quantity present throughout the section '3. Results'?

300: 'Table 5.3' does not exist!

305: 'Table 5.5' does not exist!

Author Response

Reviewer one:

Those are mine remarks and questions pointing out the weaknesses of the manuscript.

Authors Response: We would like to thank the reviewer for their time reviewing our paper and for their useful suggestions to improve the readability of the paper which we have addressed below.

27: GPS or GNSS?

Authors Response: GPS is a satellite navigation system developed by the United States government for the US Department of Defence. GLONASS is a satellite navigation system developed by the Russian Aerospace Defence Forces. GPS, GLONASS, and a handful of other navigation systems that are currently being developed, all fall under the GNSS (Global Navigation Satellite System) umbrella. When using a GPS device, you are only accessing GPS satellites; when using a GNSS device, you are accessing satellites from multiple constellations in orbit for greatly enhanced accuracy and reliability of information. As the Catapult Vector S7 is utilising GPS & GLONASS (https://support.catapultsports.com/hc/en-us/articles/360000919456-Vector-Device-Overview-S7-G7), it is considered a GNSS device. It is therefore referred to as a GNSS device throughout the paper. However, as GPS is the most commonly used navigation system in the world (most people refer to the GPS satellite constellation when talking about positioning), the term GPS is added to the keywords, and is considered an important keyword for others to find our paper.

405-406: I am not sure if a publication submitted for publication may be used as a valid reference.
99-102: Mentioning of new terms and techniques by referencing to a paper submitted for publication seems unacceptable.
129-130: Table 1 gives very few information about processing methods, their specification and algorithm properties. Moreover, the reader is advised to look for additional information in an unpublished paper.

Authors Response: Thank you for pointing this out. In response to these three comments, we have revised the manuscript so that all used methods and techniques are now clearly described in this paper to improve the readability of the paper. We have removed all references to the other unpublished manuscript that is under review. Additionally, more detail has been included in Table 1, summarizing the processing methods. We have added the following information to the methods (L120-137):

“The raw (not smoothed in any way by the manufacturer software) GNSS Doppler-shift speed data were exported and retrieved from the Catapult software files folder. The raw acceleration dataset was calculated using a central difference method on the raw Doppler-shift speed data. To determine the most appropriate custom processing method, several common smoothing methods (Butterworth filter: cut-off frequencies 0.1 to 4.9 Hz, exponential smoothing: smoothing constant 0.1 to 0.9, moving average: sliding window 0.1 s to 0.9 s) have been applied on the raw GNSS Doppler-shift speed data and were compared with a gold standard motion analysis system (Vicon) dataset. The fourth order (zero lag) low pass Butterworth filter with a cut-off frequency of 2 Hz showed the strongest relationship with the Vicon data (mean bias 0.00 m·s-2, 95% LoA ±1.55 m·s-2, RMSE 0.79 m·s-2), and was therefore used on the raw GNSS Doppler-shift speed data for the custom processing method. After using the Butterworth filter, acceleration was calculated using a central difference method on the custom processed GNSS doppler-shift speed data. Applying the processing on the raw Doppler-shift speed data before deriving acceleration data will ensure that any noise present in the raw Doppler-shift speed data will not be increased due to deriving acceleration. The manufacturer processed GNSS distance and acceleration data was exported from the manufacturer’s software using their default settings (Openfield, version 2.7.1, Catapult Sports, Melbourne, Australia).”

Furthermore, the following text has been added to the introduction to introduce the concept of custom processing, without referring to the other unpublished paper (L71-83):

“Practitioners and researchers experienced with data processing techniques might choose to extract and process the raw (not smoothed in any way by the manufacturer software) data from the GNSS devices and analyse it separately [8,19]. This approach offers several advantages, such as eliminating undesired processing practices (e.g., smoothing and algorithms) and incorporating custom processes such as new summary metrics [8,17]. Custom processing of manufacturer exported GNSS data has shown to enhance acceleration data quality, and derived summary metrics differed compared to manufacturer processed data [20]. When using custom processing on manufacturer processed GNSS data, double processing and over-smoothing of the data could take place, which could eliminate important parts of the acceleration data. Using custom processing on raw GNSS data would eliminate double processing and could enhance data quality. However, there is no research comparing custom processed raw GNSS data to manufacturer processed data.”

In section '2.3 Data Analysis': the key question is how acceleration data was obtained: by GNSS observations or by accelerometer measurements??? If the acceleration was obtained by accelerometer measurements, what the acceleration data has to do with GNSS??? If the acceleration was derived by GNSS observations, there is no word about methodology.

Authors Response: The data used in this study is GNSS acceleration data. We have added the following information to the introduction (L32-39) describing the difference between GNSS acceleration data and accelerometer-derived acceleration data:

“A GNSS device accesses satellites from multiple constellations in orbit (e.g., GPS and GLONASS)  to determine its position in space, allowing the measurement of its distance covered, speed, and acceleration [3,4]. In addition, some athlete tracking systems also include a triaxial accelerometer, gyroscope and magnetometer, allowing human activity recognition and the measurement of variables such as PlayerLoad™ [5,6]. The accelerometers within the athlete tracking system are not involved in the calculation of GNSS acceleration, accelerometer-derived acceleration is distinctly different data.”

We have enhanced the clarity in the methods stating that the used acceleration data is GNSS derived (L120-137).

115: According to specifications of the Vector S7 device (available at https://support.catapultsports.com/hc/en-us/article_attachments/360007134595) it is about 'an athlete tracking device with the capability of capturing global (GPS) and local (LPS) positioning information, inertial and heart rate data'. Therefore, presenting a tracking device as GNSS device could lead to a misleading information. I suggest to remove the term 'GNSS' from the title, because the paper itself has a little to do with GNSS and by this creating a misleading situation for the reader.

Authors Response: All data that is analysed for this study is collected using a GNSS device. The provided link of the Vector S7 indicates on page 19 (Device Specs), that the Vector S7 utilises multiple satellite constellations, and is therefore classified as a GNSS device. We have also contacted Catapult Sports (manufacturer of the S7 device) to confirm the Vector S7 is a GNSS device. We consider it important for readers to know that the athlete tracking data was collected with a GNSS device, and not an LPS/ accelerometer/IMU/Optical tracking system. It is furthermore very common for papers using athlete tracking data, to specify in their title what type of athlete tracking system was used in their paper. See the below links for some example references. We have therefore kept the term ‘GNSS’ in the title.

https://doi.org/10.3390/s23031227 - Predicting Injuries in Football Based on Data Collected from GPS-Based Wearable Sensors.
https://doi.org/10.1371/journal.pone.0260363 - A GNSS-based method to define athlete manoeuvrability in field-based team sports.
https://doi.org/10.1519/JSC.0000000000004781 - Practitioner Usage, Applications, and Understanding of Wearable GPS and Accelerometer Technology in Team Sports.

122: What is the meaning of 'CV'?

Authors Response: We have changed CV to coefficient of variation (L116-118).

163-164: Additional information and appropriate references related to 'A negative binomial generalized linear mixed model, and a linear mixed model' are missing.

Authors Response: We have added the following information to the statistical analysis (L173-175) describing why the models were chosen and included references to support both models:

“linear mixed models were used to account for recurring measures. A negative binomial generalised linear mixed model was used for the effort model, as the efforts were count based not normally distributed data [25], and a linear mixed model for the distance model [26].”

What is the meaning of 'CI' quantity present throughout the section '3. Results'?

Authors Response: The abbreviation CI (confidence interval) has now been clarified in the manuscript in L181 & L206.

300: 'Table 5.3' does not exist!

Authors Response: Thank you for pointing this out, Table 5.3 has been corrected to read Table 3 in L310.

305: 'Table 5.5' does not exist!

Authors Response: Table 5.5 has been corrected to read Table 5 in L315.

Reviewer 2 Report

Comments and Suggestions for Authors

The research analyses the impact of data processing on GNSS-based metrics to quantify team sports athlete performance.

It is an interesting study with an impact on experimental research and field application. It is well written and has a few aspects to point out.

However, I consider it unclear what raw data is being collected and how it is used for the different metrics.

Usually, these devices combine GPS/GNSS with a 3-axial accelerometer and sometimes also include a 3-axial gyroscope. Also, usually, it combines raw data that is being collected at different frequencies. The paper only mentions that GNSS tracking is being obtained at 10 Hz. But what about acceleration? Or does the device not have an accelerometer? So how is acceleration being collected? Both methods and abstract, mention that distance, speed, and acceleration are being collected, but based on what raw data? Lacks information.

Were the GNSS data and accelerometer data synchronized? to obtain the distance covered by accelerating or decelerating? Or it was done some other way? Is not clear for the reader.

Units should be separated from the numbers except % - ex. L181 - 7.5m

I suggest considering for discussion and conclusion a statement regarding the possibility of research studies using manufacturers processing data routines and analysing results that do not correspond to reality. It seems weird that authors only focus on coaches and practitioners (L23, L363).

See sentence L367-369, ...where possible.

Author Response

Reviewer two:

The research analyses the impact of data processing on GNSS-based metrics to quantify team sports athlete performance.

It is an interesting study with an impact on experimental research and field application. It is well written and has a few aspects to point out.

Authors Response: We would like to thank the reviewer for their time reviewing our paper and for their useful suggestions to improve the readability of the paper which we have addressed below.

However, I consider it unclear what raw data is being collected and how it is used for the different metrics.

Authors Response: The following text has been added to the methods section, providing a detailed description of what the raw data is and how it is being used in L120-137:

“The raw (not smoothed in any way by the manufacturer software) GNSS Doppler-shift speed data were exported and retrieved from the Catapult software files folder. The raw acceleration dataset was calculated using a central difference method on the raw Doppler-shift speed data. To determine the most appropriate custom processing method, several common smoothing methods (Butterworth filter: cut-off frequencies 0.1 to 4.9 Hz, exponential smoothing: smoothing constant 0.1 to 0.9, moving average: sliding window 0.1 s to 0.9 s) have been applied on the raw GNSS Doppler-shift speed data and were compared with a gold standard motion analysis system (Vicon) dataset. The fourth order (zero lag) low pass Butterworth filter with a cut-off frequency of 2 Hz showed the strongest relationship with the Vicon data (mean bias 0.00 m·s-2, 95% LoA ±1.55 m·s-2, RMSE 0.79 m·s-2), and was therefore used on the raw GNSS Doppler-shift speed data for the custom processing method. After using the Butterworth filter, acceleration was calculated using a central difference method on the custom processed GNSS doppler-shift speed data. Applying the processing on the raw Doppler-shift speed data before deriving acceleration data will ensure that any noise present in the raw Doppler-shift speed data will not be increased due to deriving acceleration. The manufacturer processed GNSS distance and acceleration data was exported from the manufacturer’s software using their default settings (Openfield, version 2.7.1, Catapult Sports, Melbourne, Australia).”

Usually, these devices combine GPS/GNSS with a 3-axial accelerometer and sometimes also include a 3-axial gyroscope. Also, usually, it combines raw data that is being collected at different frequencies. The paper only mentions that GNSS tracking is being obtained at 10 Hz. But what about acceleration? Or does the device not have an accelerometer? So how is acceleration being collected? Both methods and abstract, mention that distance, speed, and acceleration are being collected, but based on what raw data? Lacks information.
Were the GNSS data and accelerometer data synchronized? to obtain the distance covered by accelerating or decelerating? Or it was done some other way? Is not clear for the reader.

Authors Response: The data used in this study is GNSS acceleration data. We have added the following information to the introduction (L32-39) describing the difference between GNSS acceleration data and accelerometer-derived acceleration data:

“A GNSS device accesses satellites from multiple constellations in orbit (e.g., GPS and GLONASS)  to determine its position in space, allowing the measurement of its distance covered, speed, and acceleration [3,4]. In addition, some athlete tracking systems also include a triaxial accelerometer, gyroscope and magnetometer, allowing human activity recognition and the measurement of variables such as PlayerLoad™ [5,6]. The accelerometers within the athlete tracking system are not involved in the calculation of GNSS acceleration, accelerometer-derived acceleration is distinctly different data.”

The methods section now also includes a clear description that the used acceleration data is GNSS derived (L120-137). Further information is provided in the methods section, stating what distance and speed data is used (L120-137). Section 2.3, data analysis, now includes information describing what data is used for the distance covered metric (L166-167):

“The second metric was the distance covered in meters using the manufacturer processed GNSS distance within a predefined high (≥ 3 m·s-2 for acceleration and ≤-3 m·s-2 for deceleration) and medium (2 to 3 m·s-2 for acceleration and -2 to -3 m·s-2 for deceleration) intensity zone, extracted from each of the three acceleration datasets.”

Units should be separated from the numbers except % - ex. L181 - 7.5m

Authors Response: This has been addressed throughout the document (L206-210, 233, 262).

I suggest considering for discussion and conclusion a statement regarding the possibility of research studies using manufacturers processing data routines and analysing results that do not correspond to reality. It seems weird that authors only focus on coaches and practitioners (L23, L363).
See sentence L367-369, ...where possible.

Authors Response: We have included a discussion point on researchers using and analysing manufacturer processed data, highlighting that processing methods can change and that these changes could affect their data in lines L274-275, L282-284:

“Furthermore, researchers could be using and analysing GNSS manufacturer processed acceleration data and metrics for their research studies. Coaches, practitioners and researchers should be aware that processing methods can change, and that these changes could affect metric outputs and alter the demands they are quantifying.”

We have also extended the focus on coaches and practitioners to include researchers in the abstract and conclusion (L23, 371).

Round 2

Reviewer 1 Report

Comments and Suggestions for Authors

The manuscript has been significantly improved according to recommendations and suggestions.

Author Response

Reviewer: The manuscript has been significantly improved according to recommendations and suggestions.

Authors Response: We would like to thank the reviewer for the positive feedback and for recognizing the improvements made to the manuscript. We greatly appreciate your detailed and constructive suggestions, which have significantly contributed to enhancing the quality of our work.

Reviewer 2 Report

Comments and Suggestions for Authors

Seems to me that the paper has been improved in readability, but also is more detailed.

Considering the new information some comments:

- It should be considered that measure is different from estimate. Therefore, when mentioning that the acceleration was measured, it should be indicated that it was estimated (since it was obtained by derivation). For example L11-12; L34.

- Considering the scope of the research, which focuses on the quality of data analysis, it is pertinent to elaborate in the paper why utilizing acceleration data derived from low-frequency (Hz) speed data is preferable over using raw acceleration data measured by the device's integrated 3-axis accelerometer at a higher frequency.

Author Response

Seems to me that the paper has been improved in readability, but also is more detailed.

Authors Response: We would like to thank the reviewer for the positive feedback and for recognizing the improvements made to the manuscript. We greatly appreciate your detailed and constructive suggestions, which have significantly contributed to enhancing the quality of our work.

Considering the new information some comments:

- It should be considered that measure is different from estimate. Therefore, when mentioning that the acceleration was measured, it should be indicated that it was estimated (since it was obtained by derivation). For example L11-12; L34.

Authors Response: Thank you for pointing this out. We have amended the manuscript, indicating that acceleration was estimated rather than measured in L11, L34, L61, L111, L345.

- Considering the scope of the research, which focuses on the quality of data analysis, it is pertinent to elaborate in the paper why utilizing acceleration data derived from low-frequency (Hz) speed data is preferable over using raw acceleration data measured by the device's integrated 3-axis accelerometer at a higher frequency.

Authors Response: To make this research practical, we have chosen to use GNSS derived acceleration data rather than acceleration data measured by an accelerometer, as most researchers and practitioners (79%) in team sports use GNSS derived acceleration data. This was a finding from our scoping review (https://doi.org/https://doi.org/10.1080/02640414.2022.2054535), which mapped the athlete tracking technologies used to measure acceleration data in team sports from 2010 until 2020. To address the comment, we have added the following text to the introduction section, providing a reason to why GNSS derived acceleration data was used in L39-40: “Notably, most researchers and practitioners (79%) in team sports use GNSS derived acceleration data [7].” Followed by an explanation after the aims, describing why GNSS derived acceleration data was used instead of acceleration data measured by an accelerometer in L100-104: “To make this research practical, it was chosen to use GNSS derived acceleration data, rather than acceleration data measured by an accelerometer, as most researchers and practitioners in team sports use GNSS derived acceleration data”.